# Modulation Effects of *Toxoplasma gondii* Histone H2A1 on Murine Macrophages and Encapsulation with Polymer as a Vaccine Candidate

**DOI:** 10.3390/vaccines8040731

**Published:** 2020-12-03

**Authors:** Zhengqing Yu, Tianyuan Zhou, Yanxin Luo, Lu Dong, Chunjing Li, Junlong Liu, Jianxun Luo, Ruofeng Yan, Lixin Xu, Xiaokai Song, Xiangrui Li

**Affiliations:** 1MOE Joint International Research Laboratory of Animal Health and Food Safety, College of Veterinary Medicine, Nanjing Agricultural University, Nanjing 210095, China; 2018207044@njau.edu.cn (Z.Y.); 2018107049@njau.edu.cn (T.Z.); 14117324@njau.edu.cn (Y.L.); 18115131@njau.edu.cn (L.D.); 2017107054@njau.edu.cn (C.L.); yanruofeng@njau.edu.cn (R.Y.); xulixin@njau.edu.cn (L.X.); songxiaokai@njau.edu.cn (X.S.); 2State Key Laboratory of Veterinary Etiological Biology, Key Laboratory of Veterinary Parasitology of Gansu Province, Lanzhou Veterinary Research Institute, Chinese Academy of Agricultural Sciences, Lanzhou 730046, China; liujunlong@caas.cn (J.L.); luojianxun@caas.cn (J.L.)

**Keywords:** *Toxoplasma gondii*, H2A1, murine macrophage, nanoparticles, PLGA, chitosan, immune response

## Abstract

*Toxoplasma gondii* (*T. gondii*) is the most common zoonotic protozoa and has infected about one-third of the population worldwide. Recombinant epitopes encapsulated in nanospheres have advantages over traditional *T. gondii* vaccines. For an efficient delivery system, poly (DL-lactide-co-glycolide) (PLGA) and chitosan are the most frequently used biodegradable polymeric nanospheres with strong safety profiles. In the present study, we first expressed and purified histone H2A1 of *T. gondii* using the prokaryotic expression system. The effects of recombinant TgH2A1 on the functions of murine macrophages were then studied. Purified recombinant TgH2A1 was then encapsulated in nanospheres with PLGA and chitosan. After subcutaneous vaccination in mice, the immune response was evaluated by double antibody sandwich ELISA kits. The results from this study showed that PLGA and chitosan loaded with rTgH2A1 could trigger a stronger Th1 oriented immune response and prolong the survival time of mice effectively. In conclusion, PLGA and chitosan nanospheres loaded with histone H2A1 are an effective method for the development of vaccines against *T. gondii*. Further studies should focus on evaluating the regulatory mechanism of TgH2A1, vaccine potency, and cellular response in chronic *T. gondii* infections.

## 1. Introduction

*Toxoplasma gondii* (*T. gondii*) is an obligate intracellular protozoan parasite that causes infections in approximately 30% of the population [1,2]. It is generally asymptomatic in adults, but most infections result in severe symptoms in immunocompromised individuals [3]. Severe damage, including ocular, neurological, systemic disorders in fetuses, abortion during pregnancy, and encephalitis, can be induced by toxoplasmosis [4,5]. Human infections occur mainly through ingestion of uncooked raw meat, vegetables, and water contaminated with the oocysts of *T. gondii* [6,7,8]. For many years, drug treatments such as sulfonamides and folic acid derivatives have been unable to control this disease completely and show significant side effects [9], because the bradyzoites of *T. gondii* have a strong resistance to the environment [10].

Infected with *T. gondii*, hosts must elicit a strong immune response to suppress the replication of parasites and prevent mortality. An inhibition of *T. gondii* infection is associated with the release of interferon (IFN) γ (IFN-γ) in many cell types. The effector cells of nonhemopoietic and hemopoietic origin can be activated by IFN-γ for *T. gondii* resistance [11]. Interestingly, macrophages pre-exposed to IFN-γ can cause anti-*T. gondii* effects, but when first pre-exposed to *T. gondii* and then exposed to IFN-γ, the macrophages cannot limit the growth of *T. gondii* [12]. Furthermore, the secretion of nitric oxide (NO) plays an important role in intracellular *T. gondii* resistance [13]. According to the previous studies, secretion of NO and its intermediates plays an important role in the IFN-γ activated murine macrophages resisting intracellular duplication of *T. gondii* [11,14]. Acting as a regulating factor in L-arginine mediated NO production, inducible nitric oxide synthase (iNOS) could be enhanced by IFN-γ, leading to the inhibition of *T. gondii*. However, owing to a lack of nicotinamide adenine dinucleotide phosphate (NADPH) oxidase, the immature macrophages decrease the level of reactive oxygen species (ROS) leading to a reduced ability to inhibit the replication of *T. gondii* [15].

Concerning the immune response against *T. gondii* infections, previous research revealed that macrophages play an important role in antimicrobial activities [16]. Through a major histocompatibility complex (MHC) class II restricted antigen processing, *T. gondii* antigens stimulate macrophages and other immune cells to generate high levels of proinflammatory cytokines [17,18]. The activated macrophages mainly have two different phenotypes: classically activated macrophages (M1) and alternatively macrophages (M2) [19,20]. In resistance to *T. gondii*, the M1 macrophages are characterized by their ability to produce proinflammatory cytokines such as interleukin (IL) 1β (IL-1β), IL-12, IL-18, IL-23, tumor necrosis factor (TNF) α (TNF-α), and IFN-γ [21,22]. Unlike M1 macrophages, M2 macrophages have modulator activities to induce the production of anti-inflammatory cytokines such as IL-4, IL-10, and transforming growth factor (TGF) β (TGF-β) [23,24]. Those cytokines are necessary to a variety of processes including pathogenesis, metabolic processes, and repair mechanisms [25,26]. The regulatory mechanisms of cytokines, such as IL-10 and TGF-β, are modulated by parasite development stage, infection degree, and even host genetics [27]. Furthermore, TGF-β is also involved both in inhibition of cell proliferation and stimulation of cell apoptosis of numerous immune cells [28].

In eukaryotic cells, genomic DNA twined with histone octamers is packaged in chromatin. Histone proteins are promising targets for the development of chemotherapeutics against pathogenic parasites, because histones are basic and conserved proteins that regulate access to information contained in DNA [29]. Four core histones, including H2A, H2B, H3, and H4, have been identified to date in *T. gondii* [30], and the histone proteins play an essential role in gene expression, DNA replication, and DNA repair [31]. Histones of *T. gondii* are also involved in epigenetic gene regulation, and H2AX, which is a variant of H2A, can induce the generation of bradyzoites in vitro [32]. Among all the core histone proteins, H2A owned the most variants and engaged in DNA repair and transcription regulation. According to a previous report, mono ubiquitylation of H2A and phosphorylation of H2AX can lead to DNA damage [33].

Herein, *T. gondii* H2A1 (TgH2A1) was first expressed in a prokaryotic expression system to produce recombinant TgH2A1 (rTgH2A1). Then the effects of rTgH2A1 on modulating murine macrophage activities were investigated in vitro. Moreover, immunomodulation of rTgH2A1 encapsulated in biodegradable poly (DL-lactide-co-glycolide) (PLGA) and chitosan nanospheres have been evaluated in the mice model.

## 2. Materials and Methods

### 2.1. Ethical Statements

This study related to the animals in our research was followed by the guidelines of the Animal Ethics Committee, Nanjing Agricultural University, China. The approval ID is PZ2019080.

### 2.2. Animals and Parasites

The male institute of cancer research (ICR) mice weighing 18–22 g and female Sprague Dawley (SD) rats weighing 200–220 g were supplied by the Center of Comparative Medicine, Yangzhou University (Yangzhou, China). All the rats and mice were raised in specific pathogen-free conditions. The *T. gondii* RH strain (Type I) was supplied by the MOE Joint International Research Laboratory of Animal Health and Food Safety, College of Veterinary Medicine, Nanjing Agricultural University, Nanjing, Jiangsu, PR China. The ICR mice were used to keep *T. gondii* active by artificially challenging and collecting from the peritoneal cavity as described previously [34].

### 2.3. Cell Isolation and Culture

Obtained by immortalization of murine macrophages from C57BL/6 mice [35], the cell line Ana-1 (Institute of Cell Biology, Chinese Academy Sciences, Shanghai, China) was grown and maintained in Dulbecco’s modified Eagle’s medium (DMEM, Gibco, Beijing, China) containing 10% fetal bovine serum (FBS), 100 U/mL penicillin, and 100 μg/mL streptomycin at 37 °C in a 5% CO_2_ atmosphere.

### 2.4. Construction of the Prokaryotic Expression Plasmid

According to the guidelines of the manufacturer, Trizol reagent (Invitrogen, Shanghai, China) was used to extract the total RNA from tachyzoites of *T. gondii*. Then using the cDNA Kit (Takara Biotechnology, Dalian, China), the cDNA was constructed following the instructions. The cDNA was stored in several aliquots at −80 °C until use. Primers were designed and synthesized according to the nucleotide sequences of the H2A1 gene (Genbank: XM_002365227), the restriction sites (*Eco* RI and *Hind* III), and homologous sequences were added following the manufacturer’s instructions (Vazyme Biotech, Nanjing, China). The sense and antisense primers, 5′-GCTGATATCGGATCC GAATTC ATGAACCTTTTGGTTATTTCG-3′ and 5′-CTCGAGTGCGGCCGC AAGCTT CTATCTCCTTTGCAGCGC-3′, were synthesized to amplify the open reading frame (ORF) of H2A1 using the cDNA from *T. gondii* RH tachyzoites as a template. The PCR reaction was conducted using the following mixture: 1.25 U *EX Taq* DNA polymerase (Takara Biotechnology, Dalian, China), 5 µL 10 × *Ex Taq* buffer, 4 µL dNTP mixture (Mg^2+^ plus), 20 pmol of each primer, 2 ng cDNA template, and ddH_2_O to a final volume of 50 µL. The PCR amplification procedure included a total of 35 cycles, each consisting of denaturation at 95 °C for 30 s, annealing at 54 °C for 30 s, and extension at 72 °C for 60 s, and an initial preheating step at 95 °C for 5 min and a final extension step at 72 °C for 5 min. PCR amplification, targeting the complete 543 bp TgH2A1 ORF, was separated by agarose gel electrophoresis, then purified by following the guidelines of the E.Z.N.A. Gel Extraction Kit (Omega Biotech, Norcross, GA, USA). According to the manufacturer’s instructions, the insertions were successfully subcloned into the same restriction sites (*Eco* RI and *Hind* III) of the pET32a vector (Takara Biotechnology, Dalian, China) with the One Step Cloning Kit (Vazyme Biotech, Nanjing, China). After bacterial transformation, the positive recombinant plasmid was identified by double enzyme digest and sequence analysis through the Blast program online (http://www.blast.ncbi.nlm.nih.gov/blast.cgi).

### 2.5. Expression and Purification of TgH2A1 Recombinant Protein

The recombinant plasmid pET32a/H2A1 was transferred into *E. coli* BL21 (DE3; Invitrogen Biotechnology, Shanghai, China). Isopropyl-β-D-thiogalactopyranoside (IPTG; Sigma-Aldrich, Saint Louis, MO, USA) at a 1 mM concentration was used to induce protein expression according to the manufacturer’s instructions. After the OD600 of the bacterial culture reached approximately 0.6 at 37 °C, the bacteria were collected and then crushed by sonication. Purified using Ni^2+^-nitrilotriaceticacid column (Ni-NTA) following the manufacturer’s instructions (GE Healthcare, Piscataway, NJ, USA), the recombinant protein was isolated by 12% sodium dodecyl sulfate polyacrylamide gel electrophoresis (SDS-PAGE) and stained by Coomassie blue. The endotoxin removal was conducted using the ToxinEraser™ Endotoxin Removal Kit (GeneScript, Piscataway, NJ, USA), and then the ToxinSensor™ Chromogenic LAL Endotoxin Assay Kit (GeneScript, Piscataway, NJ, USA) was used to identify residual endotoxins of the purified protein. Using the Pierce™ BCA Protein Assay Kit (Thermo Scientific, Waltham, MA, USA) based on the Bradford protein assay, the purified recombinant protein was measured using bovine serum albumin (BSA) as a standard.

### 2.6. Western Blot Analysis of rTgH2A1 and Native TgH2A1

To develop polyclonal antibodies against rTgH2A1, a first injection containing 200 µg of purified rTgH2A1 protein with Freund’s complete adjuvant (Sigma-Aldrich, Saint Louis, MO, USA) at a 1:1 ratio was subcutaneously delivered into the back skin of SD rats. Immunized four times in an interval of 2 weeks, a booster dose containing 200 µg of purified rTgH2A1 protein mixed with Freund’s incomplete adjuvant (Sigma-Aldrich, Saint Louis, MO, USA) at a ratio of 1:1 was given in the same way. One week after the fourth immunization, the rats were bled via the eye socket and the sera were obtained.

To obtain the sera against *T. gondii*, the SD rats were artificially injected with 10^3^ tachyzoites of *T. gondii* RH strain through an intra-abdominal route. Twenty-one days after the challenge, the sera were separated using the same method mentioned above. The blank sera as negative controls were also isolated from the healthy rats kept in the same environment. All the sera were stored at −20 °C until use.

To verify rTgH2A1 protein presence, the purified rTgH2A1 was electrophoresed in 12% SDS-PAGE gel and electrotransferred to a polyvinylidene difluoride (PVDF) membrane (Immobilon-PSQ, Millipore, Billerica, MA, USA) through a semidry system (Trans-Blot SD Semi-Dry Transfer Cell, Bio-Rad, Hercules, CA, USA). The membrane was incubated with TBST (Tris-HCl buffer solution (TBS) containing 0.5% tween 20) mixed with 5% (*w*/*v*) skimmed milk powder for 2 h at 37 °C. After washing three times in TBST, the membrane was incubated with the diluted (1:100) sera against *T. gondii* for 2 h at 37 °C. After being washed three times with TBST, the membrane was then incubated with diluted (1:5000) secondary antibodies, horseradish peroxidase (HRP) conjugated goat antirat immunoglobulin (Ig) G (IgG, eBioscience, San Diego, CA, USA), in TBST for 1 h at 37 °C. Thereafter, the membrane was washed three times before visualization by applying newly prepared diaminobenzidine (Sigma-Aldrich, Saint Louis, MO, USA) as a chromogenic substrate after 5 min. All incubation steps were conducted under shaking conditions. A negative control was also created by using the sera from the healthy rats under the same conditions mentioned above.

*T. gondii* lysates were prepared as described previously with minor changes [36]. Briefly, tachyzoites of *T. gondii* were collected from the peritoneal washing fluid of artificially infected mice. The washing fluid was passed through a 5 µL filter membrane (Merck Millipore, Billerica, MA, USA) to remove cells and debris. After that, they were washed in phosphate buffer saline (PBS, pH 7.4), and tachyzoites were then suspended in PBS and disrupted by sonication on ice. To detect native TgH2A1 protein, the lysates were then electrophoresed in 12% SDS-PAGE gel. Western blotting was carried out following the method mentioned above using sera against rTgH2A1. A negative control was also performed by using the sera from the healthy rats.

### 2.7. Verification of rTgH2A1 Binding to Murine Macrophages

Flow cytometry analysis was used to confirm the binding capacity of rTgH2A1 with Ana-1 cells. The murine macrophages were cultured in a 12-well plate (Costar, CA, USA) at a density of 1 × 10^6^ cells/well, and incubated with PBS, his-tagged protein of pET32a and rTgH2A1 (20 μg/mL) for 1 h at 37 °C. The his-tagged protein of pET32a was reported as an irrelevant protein in a previous study [37]. After incubation, cells were rinsed three times and then suspended in PBS. The cells were then exposed to the sera against rTgH2A1 (1:100 dilutions) for 15 min at 37 °C. The second antibody, the mouse antirat IgG labeled with fluorescein isothiocyanate (FITC, 1:500 dilutions; eBioscience, San Diego, CA, USA), was added and incubated for 15 min at 37 °C. Before flow cytometry analysis, cells stained with antibodies were washed twice. Then the mean fluorescence intensity was calculated after flow cytometry.

### 2.8. Detection of Cell Proliferation

Following the methods of a previous study [37] with minor modifications, the Cell Counting Kit-8 (CCK-8, Beyotime, Shanghai, China) was used to detect the effect of rTgH2A1 on cell proliferation of Ana-1 cells. Briefly, Ana-1 cells were diluted to 5 × 10^5^ cells/well in a 96-well plate (Costar, CA, USA), incubated with PBS, his-tagged protein of pET32a, and different doses (0, 5, 10, 20, 40, and 80 μg/mL) of rTgH2A1 at 37 °C for 48 h. Then, the CCK-8 reagents were added into each well for 2 h according to the manufacturer’s directions. Based on the absorbance values at 450 nm (OD450), cell proliferation was quantified. Each experiment was repeated in triplicate.

### 2.9. Cell Apoptosis Assay

Murine Ana-1 cells were preincubated in a 12-well plate at a density of 5 × 10^6^ cells/well with different doses (0, 5, 10, 20, 40, and 80 μg/mL) of rTgH2A1, PBS, and his-tagged protein at 37 °C for 48 h. After that, Ana-1 cells were subsequently stained with Annexin V-FITC and propidium iodide (PI) from the Apoptosis Detection Kit (Miltenyi Biotec, Bergisch Gladbach, Germany) and analyzed by flow cytometry. Each experiment was repeated in triplicate.

### 2.10. Internalization of FITC-Dextran

Flow cytometry analysis was used to demonstrate the effects of rTgH2A1 on the phagocytic ability of murine Ana-1 cells. Diluted to 5 × 10^6^ cells/well in a 12-well plate, macrophages were then incubated with rTgH2A1 at different concentrations (0, 5, 10, 20, 40, and 80 μg/mL), PBS, and his-tagged protein for 48 h at 37 °C. Cells were washed three times and then collected in PBS after incubation with FITC-dextran (1 mg/mL in PBS) for 1 h at 37 °C. After being washed three times in PBS, the internalization of FITC-dextran was presented in median fluorescence intensity (MFI) after flow cytometry. Each experiment was repeated in triplicate.

### 2.11. Detection of NO and Cytokines Concentration in Cell Supernatants

The cell supernatants were harvested after incubation as described in Section 2.9. Based on the Griess assays [38], the production of NO was determined using the total nitric oxide assay kit (Beyotime, Shanghai, China). Briefly, cell supernatant (50 μL/well) was added to a 96-well plate with the standard (known concentration of NaNO_2_) provided by the kit, then Griess Reagent I (50 μL/well) and II (50 μL/well) were added in turn. The standard material was diluted by the same cell culture medium used for culturing Ana-1 cells. Based on absorbance values at 540 nm (OD540), the production of NO was quantified. Following the guidelines of the double antibody sandwich ELISA kit (Jinyibai, Nanjing, China), TNF-α, IL-1β, TGF-β1, and IL-10 were quantified. Based on known concentrations of mouse recombinant TNF-α, IL-1β, TGF-β1, and IL-10, the standard curves were constructed. Each independent experiment was repeated in triplicate.

### 2.12. Fabrication of PLGA Nanoparticles

The double emulsion solvent evaporation technique (*w*/*o*/*w*) was conducted using surfactant: polyvinyl alcohol (PVA, molecular weight 31,000–50,000, Sigma-Aldrich, Saint Louis, MO, USA) as described by McCall and Sirianni [39] with modifications. Briefly, 4 mg purified rTgH2A1 protein (1 mg/mL) was emulsified in 5% PLGA (molecular weight 40,000–75,000, LA/GA ratio 65/35, Sigma-Aldrich, Saint Louis, MO, USA), dissolved in 1 mL dichloromethane (DCM, Sigma-Aldrich, Saint Louis, MO, USA). Tip sonication of the first emulsion was carried out in a continuous mode for 5 s at 5 s intervals (4 min in total) under high power (20 W with 40% amplitude) on ice. Using the same permanent method as described in the first emulsion, the second emulsion was then sonicated after 5 mL PVA (6 g dissolved in 100 mL double distilled water) was dripped in to get the final *w*/*o*/*w* emulsion. Continuous stirring at 400 rpm on a magnetic stirrer at room temperature for 4 h was done to evaporate DCM. Centrifuged at 40,000 rpm for 30 min at 4 °C, particles were then washed and dissolved in double-distilled water. Stored at −80 °C for at least 1 h, the uniform suspension was then immediately lyophilized for at least 24 h. The blank nanoparticles were also prepared following the method mentioned above, except for the addition of protein rTgH2A1. All the obtained nanoparticles were stored at −80 °C until use.

### 2.13. Preparation of Chitosan Microspheres

As described previously [40], the ionic gelation technique was performed with minor modifications. Briefly, stirring at 400 rpm on a magnetic stirrer at room temperature, chitosan solution (2 mg/mL) was prepared by dissolving chitosan (molecular weight 50,000–190,000, Sigma-Aldrich, Saint Louis, MO, USA) in 1% aqueous glacial acetic acid. The pH of the resulting solution was then adjusted to 5.0 using 2 mol/L aqueous sodium hydroxide solution. Sodium tripolyphosphate (TPP, Aladdin, Shanghai, China) was dissolved in double-distilled water at a concentration of 2 mg/mL. With the magnetic stirrer stirring at 400 rpm, a 20 mL chitosan solution in a container was preheated in a water bath at 30 °C. Then, 4 mL TPP solution was quickly added to the chitosan solution. After that, 4 mg of purified rTgH2A1 protein (1 mg/mL) was dripped in at room temperature. Then, tip sonication was conducted in a continuous mode for 4 s at 3 s intervals (5 min in total) under high power (20 W with 40% amplitude) on ice. After centrifugation at 40,000 for 20 min at 4 °C, the obtained microspheres were then washed and dissolved in double-distilled water. The mixed solution was finally dried using the method mentioned in Section 2.12. The blank chitosan microspheres were also prepared using the same process except for addition of the protein rTgH2A1. All the prepared microspheres were stored at −80 °C until use.

### 2.14. Encapsulation Efficiency and Physical Characterization of Nanoparticles

A total of 4 mg of purified recombinant TgH2A1 protein was generated by nanoparticles as described above. After ultracentrifugation, the non-bound rTgH2A1 concentration in the supernatant was determined using the Pierce™ BCA Protein Assay Kit (Thermo Scientific, Waltham, MA, USA). The encapsulation efficiency of the prepared nanoparticles was calculated by applying the following formula:$$\mathrm{Encapsulation}\;\mathrm{efficiency}\;\left(\%\right)=\frac{\mathrm{Total}\;\mathrm{rTgH}2A1-\mathrm{free}\;\mathrm{rTgH}2A1}{\mathrm{Total}\;\mathrm{rTgH}2A1}\times 100\%$$

The scanning electron microscope (SEM, SU8010, Hitachi, Tokyo, Japan) at Nanjing Agricultural University was used to determine the morphology of the nanoparticles, and the diameter was determined by randomly measuring 10 nanoparticles in the SEM photographs using ImageJ software (version 1.8, NIH Image, Bethesda, MD, USA).

### 2.15. Mice Immunization and Challenge

To verify the immunogenicity of nanoparticles, the ICR mice were randomly divided into seven groups (15/group); these included the blank group, his-tagged protein group, rTgH2A1 group, PLGA group, chitosan group, rTgH2A1/PLGA group, and rTgH2A1/chitosan group. Before vaccination, 100 μg of purified rTgH2A1, freeze-dried nanoparticles encapsulated with 100 μg rTgH2A1, and 100 μg blank nanoparticles were diluted and suspended in 100 μL PBS. All groups were hypodermically injected with 100 μL/mouse at weeks 0 and 2. Blood samples were collected from five mice in each group from the eye socket on the day before each immunization, and 2 weeks after the last immunization the sera were immediately separated and stored at −20 °C until use. The remaining ten mice in each group were challenged by intraperitoneal injection with a lethal dose of *T. gondii* RH strain (200 tachyzoites) two weeks after the last immunization, then the survival time of each mouse was recorded daily.

### 2.16. Determination of Antibodies and Cytokines

Serum samples were collected from the mice at weeks 0, 2, and 4. A standard ELISA assay was conducted to determine the levels of *T. gondii* specific IgG present in the serum samples as described previously [41]. The levels of IgG1 and IgG2a, and cytokine production levels including IFN-γ, IL-4, IL-10, and IL-17, were measured using the double antibody sandwich ELISA kits (Jinyibai, Nanjing, China), referenced to known concentrations of recombinant mouse IgG1, IgG2a, IFN-γ, IL-4, IL-10, and IL-17. Each sample was analyzed in three independent experiments.

### 2.17. Statistical Analysis

Statistical analysis was conducted by using GraphPad 6.0 software (GraphPad Prism, San Diego, CA, USA). The differences between groups were identified using one-way ANOVA analysis followed by a Dunnett’s test. The survival curves were analyzed by the Kaplan–Meier test, and compared based on the log-rank/Mantel–Cox model. The fluorescence activated cell sorting (FACS) analysis was accomplished using Flowjo software (version 10, Franklin Lakes, NJ, USA). The data were expressed as means ± standard deviation. Values of *p* < 0.05 were considered significant.

## 3. Results

### 3.1. Restriction Analysis of pET32a/H2A1

Restriction analysis showed the TgH2A1 gene was correctly inserted in the plasmid and conserved restriction sites. The pET32a/H2A1 was linearized at 6430 bp by single digestion with *Eco* RI and *Hind* III, while it generated two fragments of 549 bp and 5881 bp by double digestion with the restriction enzymes mentioned above (Appendix A). The sequence analysis (Appendix A) also indicated that recombinant plasmid pET32a/H2A1 was correctly constructed.

### 3.2. Western Blot Analysis of Purified rTgH2A1

After the expression purification and endotoxin removal of recombinant protein, the endotoxin level of the purified rTgH2A1 was decreased to 0.1 EU/mL, and then rTgH2A1 was subjected to SDS-PAGE analysis yielding a band of approximately 37 kDa visualized through Coomassie blue staining (Figure 1a). Theoretically, the molecular mass of native TgH2A1 was 19.5 kDa, and the his-tagged protein was 18.0 kDa according to the expression region of the pET32a vector. Thus, the molecular mass of recombinant TgH2A1 was 37.5 kDa in theory. The results showed that the recombinant TgH2A1 was correctly expressed. Western bolt results showed that recombinant TgH2A1 reacted with sera separated from rats challenged with *T. gondii* (Figure 1b), while native TgH2A1 from *T. gondii* lysates could also be identified by anti-rTgH2A1 polyclonal antibodies (Figure 1c) at the position of 19.5 kDa. The results showed that recombinant TgH2A1 had immunogenicity and could elicit a host immune response.

### 3.3. Confirmation of the Combination of rTgH2A1 with Murine Macrophages

Flow cytometry analysis was used to verify the binding ability of rTgH2A1 with Ana-1 cells. The mean fluorescence intensity of Ana-1 macrophages incubated with rTgH2A1 (141,020.300 ± 9642.225) was significantly higher (*p* < 0.001, Figure 2), compared to the blank (23,293.67 ± 880.815) and control groups (27,437.000 ± 773.630). Furthermore, no significance was shown between the blank and control group (*p* > 0.05). The results indicate that murine macrophages could recognize and bind to rTgH2A1.

### 3.4. rTgH2A1 Promoted the Proliferation of Murine Macrophages and Induced Apoptosis

Treated with CCK-8 reagent, the cell proliferation capacity was conducted to confirm the effects of rTgH2A1. The OD450 showed that no significant difference was observed between the blank group preincubated with PBS and the control group preincubated with his-tagged protein, and the proliferation of Ana-1 cells incubated with rTgH2A1 at the concentration of 40 μg/mL was significantly promoted compared with the blank and control groups (*p* < 0.05, Figure 3). The results expressed as mean ± standard deviation including *p* values are shown in Appendix A. The amount of early- and late-stage apoptotic cells was subsequently detected using an Annexin V-FITC kit. The results showed that early-stage apoptosis was promoted by coculturing with rTgH2A1 at concentrations of 20, 40, and 80 μg/mL (*p* < 0.001, Figure 4a).

### 3.5. rTgH2A1 Increased Murine Macrophages in Phagocytosis and NO Secretion

Ana-1 cells were exposed to different concentrations of rTgH2A1 (0, 5, 10, 20, 40, and 80 μg/mL) for 48 h, and the ability of murine Ana-1 cells to engulf FITC-dextran was significantly promoted by coculturing with rTgH2A1 at the concentration of 40 μg/mL (*p* < 0.001) and 80 μg/mL (*p* < 0.05), compared to the blank and control groups (Figure 5a). After treatment with rTgH2A1, NO production was significantly stimulated in supernatants of Ana-1 macrophages at all concentrations (5, 10, 20, 40, and 80 μg/mL) of rTgH2A1 compared with the blank and control groups (*p* < 0.001, Figure 5b).

### 3.6. rTgH2A1 Modulates the Cytokines of Murine Macrophages

Cell supernatants were harvested to assess the cytokine production of murine macrophages, and no significant effects were observed between the control group treated with his-tagged protein and the blank group treated with PBS. The proinflammatory cytokines including TNF-α and IL-1β were significantly induced after being preincubated with rTgH2A1 at concentrations of 10, 20, 40, and 80 μg/mL (*p* < 0.001). Compared with the blank and control groups, 5 μg/mL rTgH2A1 could induce the secretion of TNF-α (*p* < 0.001) and IL-1β (*p* < 0.01). As for anti-inflammatory cytokines (TGF-β1 and IL-10), only 20 μg/mL rTgH2A1 promoted TGF-β1 secretions compared with the blank group (*p* < 0.01). When incubated with 80 μg/mL rTgH2A1, TGF-β1 secretions were inhibited compared with the blank and control groups (*p* < 0.001). However, rTgH2A1 had no significant effect on IL-10 production (Figure 6).

### 3.7. Physical Characterization and Encapsulation Efficiency of Nanoparticles

After encapsulation of rTgH2A1 in PLGA and chitosan microspheres, the morphology of nanoparticles was examined with scanning electron microscopy. The average diameter of rTgH2A1/PLGA microspheres was about 98 nm, while the chitosan microspheres were approximately 102 nm in average diameter (Figure 7). The encapsulation efficiency of two types of microspheres was analyzed. As the concentration of rTgH2A1 reached 1000 μg/mL, the encapsulation efficiency of rTgH2A1/PLGA microspheres reached 74.8%, while the chitosan microspheres encapsulated with rTgH2A1 reached 67.4%.

### 3.8. Humoral Response and Cytokine Production

To evaluate the level of specific IgG, sera samples were collected prior to each immunization and two weeks after the last immunization. A significantly higher level (*p* < 0.001) of total IgG was observed in the sera immunized with rTgH2A1, rTgH2A1/PLGA, and rTgH2A1/chitosan, compared with the blank and his-tagged protein groups after the first and second immunization (Table 1). Furthermore, the levels of IgG1 and IgG2a were also measured in the sera from mice two weeks after the last immunization. As displayed in Table 2, significant production of IgG1 was observed in the mice immunized with rTgH2A1/PLGA compared with the blank group (*p* < 0.001) and his-tagged protein group (*p* < 0.05). The secretions of IgG2a were significant (*p* < 0.001) in the rTgH2A1 rTgH2A1/chitosan and rTgH2A1/PLGA group compared to the blank group and his-tagged protein group. Moreover, the concentrations of IgG2a were higher compared with the level of IgG1 in the rTgH2A1/nanosphere vaccinated groups indicating that mice immunized with rTgH2A1/PLGA and rTgH2A1/chitosan could generate a Th1-biased humoral immune response.

The sera separated from each mouse in a different group at weeks 0, 2, and 4 were used to obtain the levels of IFN-γ, IL-4, IL-10, and IL-17, according to the manufacturer’s instructions from commercial kits based on the double antibody sandwich method. As shown in Figure 8a, higher levels (*p* < 0.001) of IFN-γ were detected in groups immunized with rTgH2A1/chitosan and rTgH2A1/PLGA compared with the blank and his-tagged protein groups at weeks 2 and 4. The mice immunized with rTgH2A1 could generate significantly higher IFN-γ compared with the blank group in weeks 2 and 4 (*p* < 0.001) and the his-tagged protein group in weeks 2 (*p* < 0.001) and 4 (*p* < 0.01). As shown in Figure 8b, mice immunized with rTgH2A1/chitosan and rTgH2A1/PLGA showed a significant level of IL-4 compared with the blank and his-tagged protein groups (*p* < 0.001) in week 4, while the mice immunized with rTgH2A1 produced a higher level in week 2 (*p* < 0.05). Mice immunized with rTgH2A1/PLGA could generate a significantly higher level of IL-17, compared with the blank (*p* < 0.01) and his-tagged protein groups (*p* < 0.05) in week 4 (Figure 8d). Additionally, in week 4, mice immunized with rTgH2A1 alone could secrete a higher level of IL-17 compared with the blank group (*p* < 0.05, Figure 8d). However, there were no significant changes observed in the levels of IL-10 secreted by mice among different groups (Figure 8c).

### 3.9. Microsphere Immunization against Acute Toxoplasmosis

To analyze whether the two types of nanoparticles could confer effective protection against acute toxoplasmosis, each immunized mouse was injected with 200 tachyzoites of the virulent RH strain 2 weeks after the last immunization. As illustrated in Figure 9, all mice vaccinated with PBS, his-tagged protein, PLGA, and chitosan succumbed within 12 days of infection. Compared to the blank (9.300 ± 0.900 days) and his-tagged group (9.100 ± 1.135 days), a significant prolonged survival time (*p* < 0.001) in mice vaccinated with rTgH2A1 (12.200 ± 1.326 days), rTgH2A1/chitosan (14.700 ± 1.86 days), and rTgH2A1/PLGA (13.900 ± 1.135 days) was observed. Moreover, animals vaccinated with rTgH2A1/chitosan showed a significant increase (*p* < 0.001) in their survival time with a maximum survival time of 17 days.

## 4. Discussion

In our study, we first expressed the recombinant protein TgH2A1 by the prokaryotic expression system. Purified by Ni-NTA chelating affinity chromatography, the purified rTgH2A1 affected the proliferation of murine macrophages and induced apoptosis. After coincubation with rTgH2A1, phagocytotic capacity, and NO secretion of murine Ana-1 cells were enhanced significantly, and the cytokines were also modulated. PLGA and chitosan nanospheres loaded with adequate rTgH2A1 prevented the protein from enzymatic degradation and allowed the stable release rTgH2A1. During in vivo experiments, prolonged survival was observed in mice vaccinated with nanospheres, accompanied by intensive humoral and cellular responses.

There is accumulating evidence suggesting that epigenetics and chromatin structure are crucial in parasitic development [42], and the effects of histone proteins of *T. gondii* on multiple developmental stages and complex life cycles are considered to be tremendous [43]. Almost every life process of *T. gondii*, such as growth, differentiation, cellular development, and even gene expression, is closely attached to the nucleosome, which is made up of DNA and histone proteins [32]. Some emerging studies suggested that these highly conserved histones play an important role in evading host immune responses and altering phenotypes at some key points of the life cycle [44,45]. It is worth mentioning that native TgH2A1 is not detected in mature bradyzoites from brain systems of experimental mice infected with *T. gondii* [32]. Based on this, histone proteins and related enzymes are a potential target for developing new therapeutic strategies against parasites [31,37]. When the purified rTgH2A1 was accessible, we first confirmed its combination with murine macrophages by flow cytometry methods.

Participating in the innate immune reaction, macrophages are the first immune defense response. As the target cells of the immune system escape, plastic macrophages play a key role in anti-*T. gondii* infections [46,47]. In our study, rTgH2A1 slightly promoted the proliferation of murine Ana-1 cells at the concentration of 40 μg/mL. Interestingly, contrary to a dose effect, proliferation at the concentration of 80 μg/mL was not significant compared with the blank and control groups. During incubation with rTgH2A1, early-stage apoptosis was greatly promoted at the concentration of 80 μg/mL, and this may affect some functions of Ana-1 cells leading to a low proliferation. However, in previous reports, recombinant histone 4 from *T. gondii* inhibited the proliferation and induced the apoptosis of murine macrophages in vitro [37], and similar results also obtained with excretory/secretory antigens from *T. gondii* [48]. All the results revealed that macrophages are regulated by rTgH2A1, and a suitable environment could be created for anti-*T. gondii* infection.

FITC-dextran is commonly used as a fluorescent probe to analyze cell permeability or phagocytosis. FITC-dextran can be absorbed by macrophages through C-type lectin, a mannose receptor [49,50]. Some reports have revealed that the mannose receptor mediates phagocytic activity of bacteria strains [51,52,53]. After incubation with purified rTgH2A1, the phagocytic ability of Ana-1 cells was significantly improved, and that enhanced the removal of *T. gondii*. Furthermore, the secretions of NO and cytokines are another method of eradicating pathogens by activated macrophages [54,55]. Many proinflammatory cytokines can be generated during *T. gondii* infection, but strong immune reactions could contribute to tissue damage [56]. Thus, the anti-inflammatory ability of macrophages was particularly important. In the present study, proinflammatory cytokines, anti-inflammatory cytokines, and secretion of NO were detected to analyze the modulations of rTgH2A1 on macrophages. We found that Ana-1 cells secreted more proinflammatory cytokines (TNF-α and IL-1β) and NO, and that is expected to inhibit the replication of *T. gondii* [57,58,59]. The TGF-β1 level was modulated in Ana-1 cells in response to rTgH2A1 in a dose-independent manner. The complicated role of TGF-β might be caused by the complex host–parasite interaction or the host genetic modifications, and needs further exploration

Peptide-based vaccines also have some weaknesses. With poor immunogenicity, they are easy to be enzymatically broken. Thus an effective delivery system is needed [60]. Nanocarriers can induce enhanced recognition and robust immune responses compared to the naked antigens [61]. Many types of nanospheres have been studied, and current concerns are focused on PLGA and chitosan. Trials have been completed using nanospheres in immunizations against bacterial, viral, and parasitic diseases with promising results [62,63,64]. Approved as a delivery system for drugs by the US Food and Drug Administration (USFDA), PLGA is a biodegradable polymeric nanosphere with good biocompatibility [65]. PLGA nanospheres showed no adverse effect on the functions of dendritic cells, such as migration, cytokine secretion, and costimulatory properties [66]. Furthermore, PLGA nanospheres can deliver both the antigens and immunoenhancers to macrophages or dendritic cells [67]. At present, PLGA has been used to protect peptides from undesirable degradation, for example, recombinant tachyzoite surface antigen 1, rhoptry protein 18, calcium-dependent protein kinase 6, and so on [68,69,70]. Obtained from the deacetylation of chitin, chitosan possesses unique chemical and biological properties, such as being easy to load with peptides, non-toxicity, easily removed from the body, relatively no side effects, cationic properties, biodegradability, and bioadhesive characteristics, which have been extensively studied [71,72,73]. As a modified natural carbohydrate polymer, chitosan can enhance the immunogenicity of antigens and activate the immune system [74]. Moreover, as an adjuvant, chitosan has been shown to enhance the activation of dendritic cells and the Th1 type immune response via the cyclic GMP-AMP synthase (cGAS)-stimulator of interferon genes (STING) pathway [75,76]. In our study, the rTgH2A1/PLGA nanospheres prepared using the double emulsion solvent evaporation technique (*w*/*o*/*w*) were observed to be spherical in shape and had uniform diameters around 98 nm, while the chitosan nanospheres prepared using the ionic gelation technique were also spherical in shape with an average diameter around 102 nm. It was remarkable that DCM could be used in PLGA nanospheres, which were often limited by toxicity and could not be thoroughly removed by evaporation at room temperature [77]. To reduce this problem, the PLGA particles were fully lyophilized to minimize the effect of DCM. As expected, the PLGA nanospheres showed good immune protection and almost no cytotoxicity.

Associated with inhibition of the combination of pathogens in host cells and activation of the complement pathway, it has long been recognized that humoral responses and antibodies played a critical role in immunity against *T. gondii* [78,79]. IgG2a is related to the Th1 type immune response while IgG1 is relevant to the Th2 type immune response [80]. Our results indicated that rTgH2A1/chitosan nanospheres could generate Th1 oriented immune response with more IgG2a than IgG1. Although mice vaccinated with rTgH2A1/PLGA nanospheres could generate significantly higher IgG1 and IgG2a, rTgH2A1/PLGA nanospheres could also generate Th1 oriented immune responses due to the IgG2a/IgG1 ratios. To sum up, our data lent credit to the idea that both rTgH2A1/PLGA and rTgH2A1/chitosan nanospheres could generate stronger Th1 oriented immune responses.

Some reports revealed that the administration route of nanoparticles could impact immunogenicity. Hamdy et al. reported that PLGA particles were internalized by dendritic cells through subcutaneous or intradermal routes, whereas intraperitoneal administration resulted in PLGA uptake by macrophages [67]. Furthermore, a single subcutaneous injection of HBsAg-loaded PLGA microspheres in mice can lead to antibody responses comparable to those of three injections of the HBsAg aluminum vaccine [81]. Thus, the development of effective single-dose vaccines against *T. gondii* infections that can elicit long lasting protective immunity will be the key to preventing *T. gondii* infections. In our study, we found that the single subcutaneous immunization with nanoparticles in mice resulted in a certain protection; thus, further studies should investigate the impact on other administration routes and determine the optimal dose of nanoparticles.

Mediated by CD4^+^ Th1 cells, the Th1 type immune response inhibited the replication of tachyzoites and reactivation of cysts [82]. Therefore, T cell activation plays an important role in the suppression of *T. gondii* [80]. As the cytokine of Th1 type lymphocytes, IFN-γ can induce an inflammatory response and control the replication of *T. gondii* at the early stages of infection [83]. Delivered by nanospheres, rTgH2A1 was continuously and stably released, and strong cellular immune responses were activated. In the present study, we found the mice injected with rTgH2A1 nanospheres could generate a higher level of IFN-γ compared with the mice injected with only rTgH2A1, and all the groups vaccinated with rTgH2A1 and its nanospheres could generate a higher level of IFN-γ compared with the his-tagged protein groups. Thus, we speculated that rTgH2A1 encapsulated in nanospheres could induce a stronger cellular immune response than rTgH2A1 due to the delivery system. Additionally, many reports showed similar results in nanospheres against *T. gondii* [84,85,86].

Provoked by a lethal dose of IFN-γ, the death of mice can be caused by severe inflammation, not *T. gondii* infection [82]. Produced by the Th2 type immune response, IL-4 and IL-10 could antagonize the effects of IFN-γ [87]. We also found the mice immunized with two types of nanospheres produced more IL-4 than the his-tagged protein group (week 4 in Figure 8b), but IL-10 remained similar to the his-tagged protein group. Reducing inflammatory injury and fatality of *T. gondii* infection, cytokines produced by the Th2 type immune response play an important role in prolonging the survival time of mice [88]. Our results provided excellent support for this conclusion. It should be noted that purified rTgH2A1 could not provoke significant secretion of IL-4 and IL-10, but this did not mean that the purified rTgH2A1 could trigger severe inflammation.

As an important regulator secreted by Th17 cells, CD4^+^ T cells, and NK cells in inflammation, IL-17 is also involved in *T. gondii* infection [89]. IL-17 can recruit neutrophils to infected sites [90], and IL-17A-deficient mice are susceptible to *T. gondii* infection [87]. However, according to reports in the literature, mice with signaling receptor IL-17RA deficiency show reduced ileitis and inflammation in other organs and prolonged survival [91]. Thus, the role of IL-17 in *T. gondii* infection still remained unexplored. In our study, we found that the mice vaccinated with rTgH2A1/PLGA produced a higher level of IL-17 compared with his-tagged protein groups (week 4 in Figure 8d). During *T. gondii* infection, neutrophils could be induced by IL-17 mediated signaling [90]. From this perspective, rTgH2A1 encapsulated in PLGA could induce the production of IL-17, which was involved in anti-*T. gondii*. IL-17 is worth further investigation in immune function.

The survival rate of mice is considered the best way to evaluate *T. gondii* vaccine efficiency [92]. In our study, we found that mice vaccinated with two types of nanospheres had significantly prolonged survival time after intraperitoneal injection of a lethal dose of *T. gondii*. The mice vaccinated with rTgH2A1 survived more days compared with the his-tagged protein group, but fewer days compared with the group vaccinated with two types of nanospheres. All these results indicated that rTgH2A1 encapsulated in two types of nanospheres could prolong survival of animals with acute *T. gondii* infection. However, in natural infections, animal exposure to *T. gondii* typically occurs due to the ingestion of food or drink contaminated by parasite cysts [93]. Primary infections in most adults are asymptomatic and more prone to developing chronic infection [94]. Further studies should evaluate the immune protection induced by rTgH2A1 and its nanospheres in chronic *T. gondii* infections.

## 5. Conclusions

In the present study, we first proved that the recombinant TgH2A1 was recognized by murine macrophages and triggered an array of immune responses to inhibit the replication of *T. gondii*. Then the purified TgH2A1 was encapsulated in nanospheres and vaccinated with mice, and both rTgH2A1/PLGA and rTgH2A1/chitosan nanospheres could trigger a stronger Th1 oriented immune response compared with the mice vaccinated with only rTgH2A1. Our results indicated that PLGA or chitosan nanospheres loaded with antigenic peptides could be a promising way to develop a vaccine against *T. gondii*. Future studies are necessary to evaluate the regulatory mechanism of TgH2A1, and improve the vaccine potency and determine the cellular response in chronic *T. gondii* infections.

## Figures and Tables

**Figure 1 vaccines-08-00731-f001:**
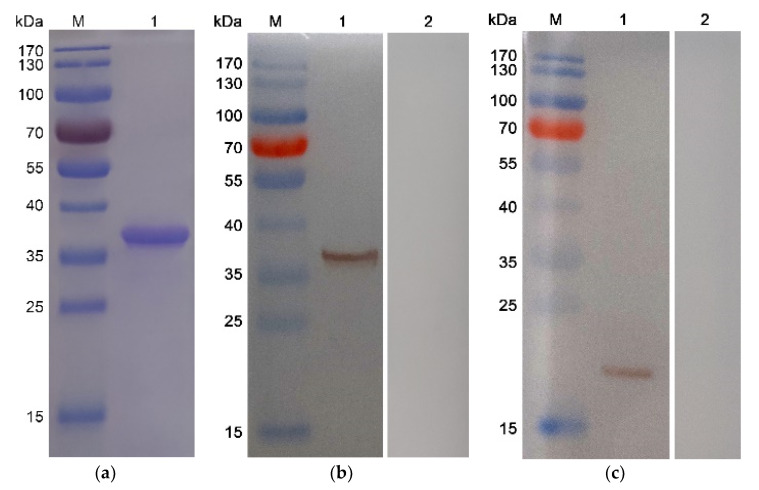
(**a**) SDS-PAGE analysis of the purification of recombinant TgH2A1. Line M: protein molecular weight marker; Line 1: rTgH2A1 purified by Ni^2+^ charged column chromatography and dialysis. (**b**) Western blotting of recombinant TgH2A1. Line M: protein molecular weight marker; Line 1: rTgH2A1 probed by serum from rats against *T. gondii* as the primary antibody; Line 2: rTgH2A1 probed by sera from normal rats as the primary antibody. (**c**) Western blotting of native TgH2A1. Line M: protein molecular weight marker; Line 1: total soluble protein of *T. gondii* tachyzoites probed by sera from rats immunized by rTgH2A1 as the primary antibody; Line 2: total soluble protein of *T. gondii* tachyzoites probed by sera of normal rats as the primary antibody.

**Figure 2 vaccines-08-00731-f002:**
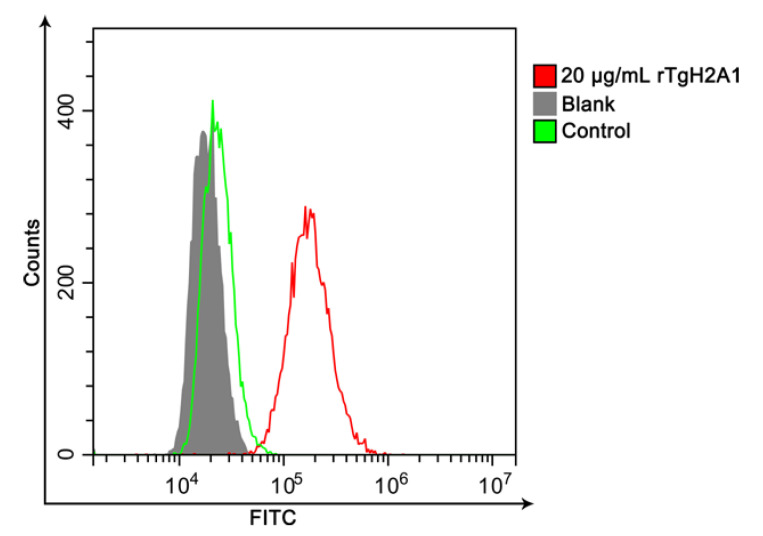
Representative FACS histograms showed a shift in fluorescence intensity of murine Ana-1 cells combined with rTgH2A1 labeled with FITC. In the blank group and control group, the Ana-1 cells were treated with PBS and purified his-tagged protein, respectively, while the experimental group was treated with purified recombinant TgH2A1 (20 μg/mL). After incubation, the mouse antirat IgG labeled with FITC was added.

**Figure 3 vaccines-08-00731-f003:**
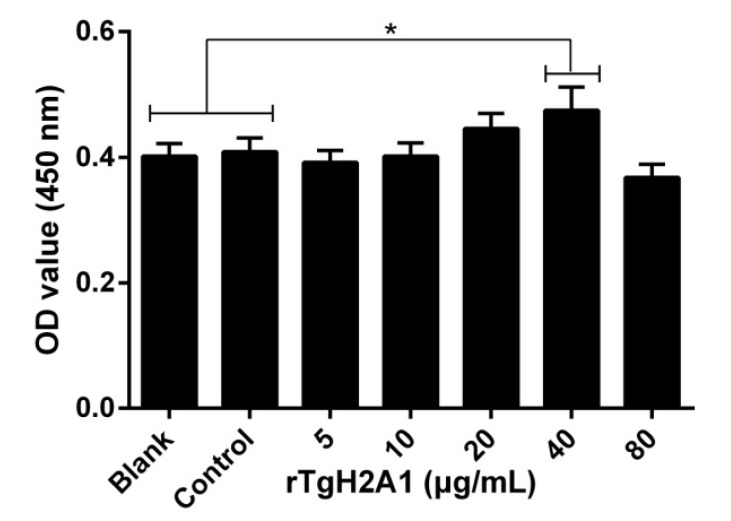
Effects on proliferation of different concentrations (0, 5, 10, 20, 40, and 80 μg/mL) of rTgH2A1 on the murine macrophages. The proliferation assay was performed using the CCK-8 reagent kit. The cell proliferation index was calculated using the OD450 values. The Ana-1 cells in the blank groups were treated with PBS while those in the control group were treated with his-tagged protein. Results were evaluated using one-way ANOVA analysis followed by Dunnett’s test and expressed as mean ± standard deviation of three independent experiments. * *p* < 0.05 compared with the blank or control group.

**Figure 4 vaccines-08-00731-f004:**
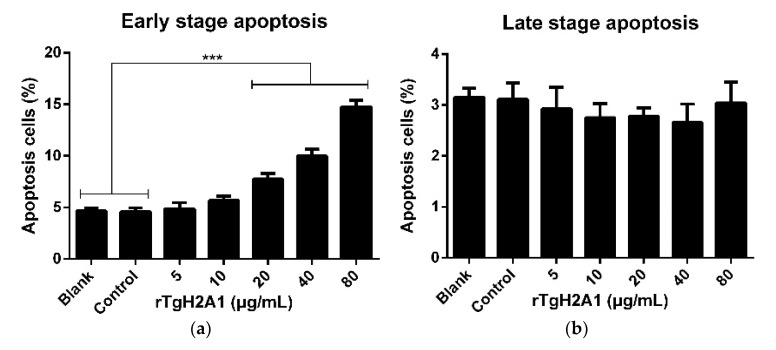
rTgH2A1 induced the apoptosis of murine Ana-1 cells. Ana-1 cells were preincubated with different concentrations (0, 5, 10, 20, 40, and 80 μg/mL) of rTgH2A1 for 48 h. Results were evaluated using one-way ANOVA analysis followed by Dunnett’s test and expressed as mean ± standard deviation of three independent experiments. *** *p* < 0.001 compared with the blank group or control group. Effect of rTgH2A1 on the early (**a**) and late (**b**) stage apoptosis of murine Ana-1 cells. Cells stained with Annexin V and PI were measured with flow cytometry (**c**).

**Figure 5 vaccines-08-00731-f005:**
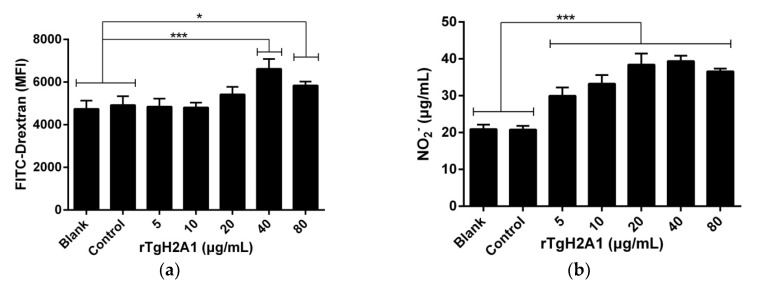
rTgH2A1 induced phagocytosis and NO secretion of the murine macrophages in vitro. Preincubated with different concentrations (0, 5, 10, 20, 40, and 80 μg/mL) of rTgH2A1 for 48 h, phagocytosis and NO secretion were respectively determined by flow cytometry and commercial kits. The Ana-1 cells in the blank groups were treated with PBS while the control group was treated with his-tagged protein. Results were evaluated using one-way ANOVA analysis followed by Dunnett’s test, and values were mean ± standard deviation of three independent experiments. * *p* < 0.05 and *** *p* < 0.001 compared with the blank group or control group. (**a**) rTgH2A1 significantly induced phagocytosis of the murine macrophages. The median fluorescence intensity (MFI) was established based on the statistical data from flow cytometry analysis. (**b**) rTgH2A1 stimulated NO production of murine Ana-1 cells. The total nitric oxide kit based on the Griess assay was used to determine NO production.

**Figure 6 vaccines-08-00731-f006:**
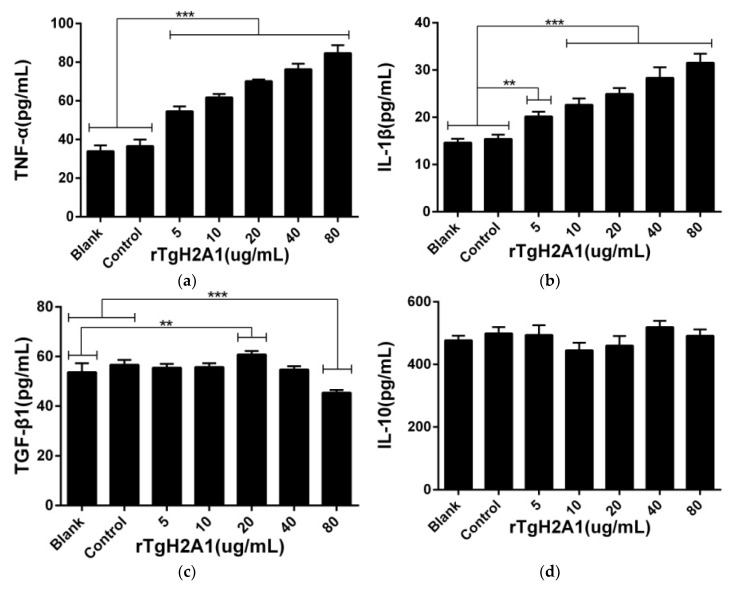
rTgH2A1 induced cytokine secretion of the murine macrophages in vitro. Preincubated with different concentrations (0, 5, 10, 20, 40, and 80 μg/mL) of rTgH2A1 for 48 h, cytokine secretion was determined by commercial kits based on the double antibody sandwich method. The Ana-1 cells in the blank groups were treated with PBS while the control group was treated with his-tagged protein. Results were evaluated using one-way ANOVA analysis followed by Dunnett’s test, and values were displayed as mean ± standard deviation from three independent experiments. ** *p* < 0.01, and *** *p* < 0.001 compared with the blank group or control group. Effect of rTgH2A1 on the TNF-α (**a**), IL-1β (**b**), TGF-β1 (**c**), and IL-10 (**d**) secretion of murine Ana-1 cells was determined.

**Figure 7 vaccines-08-00731-f007:**
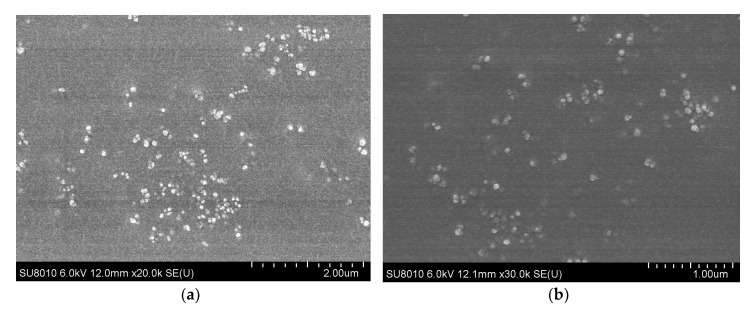
SEM images of microspheres. (**a**) Using PVA as a surfactant, rTgH2A1/PLGA microspheres were prepared using the double emulsion solvent evaporation technique (*w*/*o*/*w*). (**b**) rTgH2A1/chitosan microspheres were prepared using the ionic gelation technique.

**Figure 8 vaccines-08-00731-f008:**
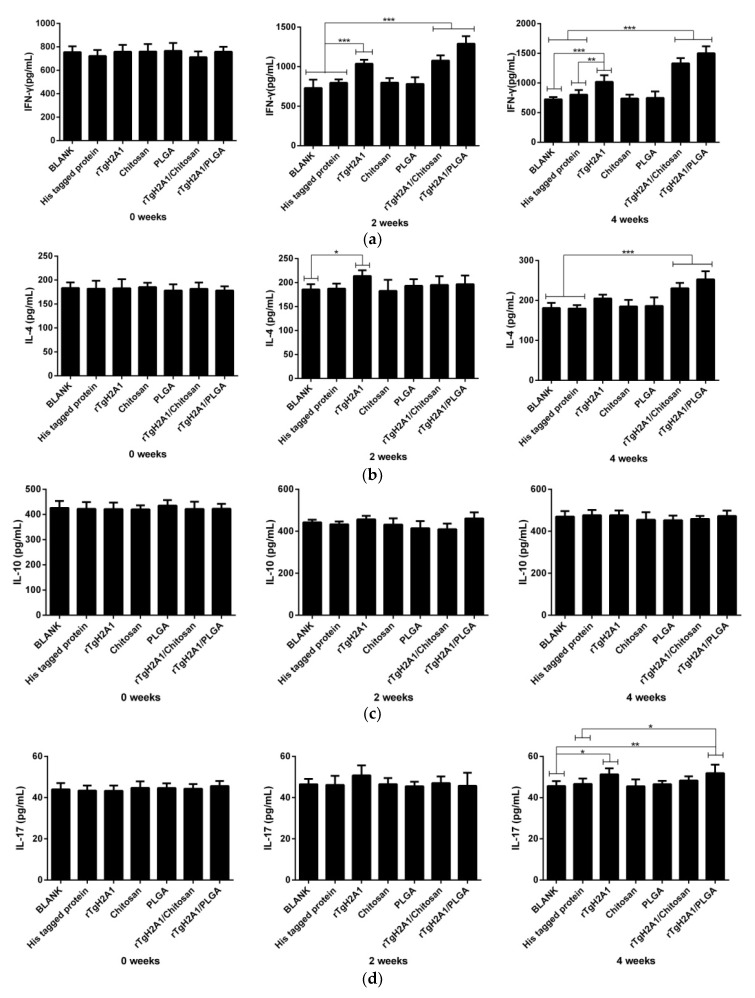
Cytokine production. Double antibody sandwich ELISA kits were used to determine the level of IFN-γ (**a**), IL-4 (**b**), IL-10 (**c**), and IL-17 (**d**) in sera from animals collected at weeks 0, 2, and 4. Results were evaluated using one-way ANOVA analysis followed by Dunnett’s test, and values were shown as mean ± standard deviation of pg/mL. * *p* < 0.05, ** *p* < 0.01, and *** *p* < 0.001 compared with the blank group or his-tagged protein group.

**Figure 9 vaccines-08-00731-f009:**
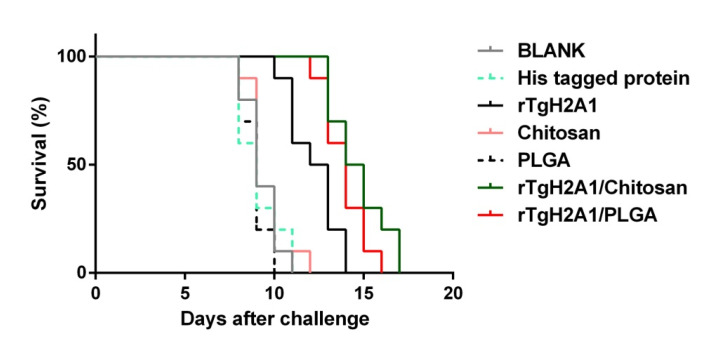
Protection of nanoparticle-immunized mice against acute *T. gondii* infection. Two weeks after the last immunization, ten mice from each group were intraperitoneally injected with 200 tachyzoites of the highly virulent *T. gondii* RH strain. Results were analyzed by the Kaplan–Meier test, and compared based on log-rank/Mantel–Cox model.

**Table 1 vaccines-08-00731-t001:** Determination of total IgG antibodies in the sera from mice immunized with PBS, his-tagged protein, rTgH2A1, PLGA, chitosan, rTgH2A1/PLGA, and rTgH2A1/chitosan on weeks 0, 2, and 4. Results were evaluated using one-way ANOVA analysis followed by Dunnett’s test and shown as the mean of the OD450 ± standard deviation.

Weeks	Group	Total IgG (OD Values)	*p* Value	*p* Value
0 weeks	Blank	0.605 ± 0.059	-	0.9965 ^b^
	His-tagged protein	0.594 ± 0.037	0.9965 ^a^	-
	rTgH2A1	0.605 ± 0.045	>0.9999 ^a^	0.9978 ^b^
	Chitosan	0.611 ± 0.031	0.9997 ^a^	0.9811 ^b^
	PLGA	0.605 ± 0.045	>0.9999 ^a^	0.9978 ^b^
	rTgH2A1/Chitosan	0.619 ± 0.015	0.9930 ^a^	0.8857 ^b^
	rTgH2A1/PLGA	0.626 ± 0.042	0.9542 ^a^	0.7542 ^b^
2 weeks	Blank	0.621 ± 0.033	-	0.7318 ^d^
	His-tagged protein	0.671 ± 0.046	0.7318 ^c^	-
	rTgH2A1	0.918 ± 0.030	<0.0001 ^c^	<0.0001 ^d^
	Chitosan	0.697 ± 0.081	0.3489 ^c^	0.9802 ^d^
	PLGA	0.633 ± 0.027	0.9996 ^c^	0.8878 ^d^
	rTgH2A1/Chitosan	1.112 ± 0.091	<0.0001 ^c^	<0.0001 ^d^
	rTgH2A1/PLGA	1.126 ± 0.085	<0.0001 ^c^	<0.0001 ^d^
4 weeks	Blank	0.648 ± 0.031	-	0.8815 ^f^
	His-tagged protein	0.688 ± 0.037	0.8815 ^e^	-
	rTgH2A1	0.998 ± 0.092	<0.0001 ^e^	<0.0001 ^f^
	Chitosan	0.693 ± 0.026	0.8187 ^e^	0.9999 ^f^
	PLGA	0.644 ± 0.049	0.9999 ^e^	0.8317 ^f^
	rTgH2A1/Chitosan	1.367 ± 0.087	<0.0001 ^e^	<0.0001 ^f^
	rTgH2A1/PLGA	1.489 ± 0.085	<0.0001 ^e^	<0.0001 ^f^

^a^, ^b^, ^c^, ^d^, ^e^ and ^f^ were compared with the blank group at 0 weeks, the his-tagged protein group at 0 weeks, the blank group at 2 weeks, the his-tagged protein group at 2 weeks, the blank group at 4 weeks and the his-tagged protein group at 4 weeks, respectively.

**Table 2 vaccines-08-00731-t002:** Determination of IgG subclasses IgG1 and IgG2a in the sera of the immunized mice two weeks after the last immunization. Results were evaluated using one-way ANOVA analysis followed by a Dunnett’s test and shown as the mean of the OD450 ± standard deviation.

Group	IgG1 (OD Values)	IgG1 Concentrations (ng/mL)	*p* Value ^a^	*p* Value ^b^	IgG2a (OD Values)	IgG2a Concentrations (ng/mL)	*p* Value ^a^	*p* Value ^b^
Blank	0.389 ± 0.024	1876.942 ± 104.595	-	0.3985	0.649 ± 0.042	1543.657 ± 116.373	-	0.9164
His-tagged protein	0.425 ± 0.022	2031.688 ± 96.355	0.3985	-	0.678 ± 0.026	1623.244 ± 73.475	0.9164	-
rTgH2A1	0.439 ± 0.021	2095.674 ± 93.427	0.1218	0.9612	0.900 ± 0.060	2260.896 ± 179.102	<0.0001	< 0.0001
Chitosan	0.400 ± 0.023	1924.944 ± 100.771	0.9906	0.7385	0.684 ± 0.030	1642.357 ± 83.402	0.8166	0.9997
PLGA	0.395 ± 0.039	1905.677 ± 168.024	0.9996	0.5971	0.643 ± 0.043	1527.466 ± 118.901	0.9997	0.8342
rTgH2A1/Chitosan	0.442 ± 0.030	2109.378 ± 133.144	0.0908	0.9121	1.120 ±0.022	2932.664 ± 70.216	<0.0001	<0.0001
rTgH2A1/PLGA	0.489 ± 0.046	2311.107 ± 201.347	0.0005	0.0304	1.206 ± 0.073	3213.309 ± 240.906	<0.0001	<0.0001

^a^ and ^b^ were compared with the blank group and his-tagged protein group, respectively.

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
