# Peer review of "Modulation Effects of Toxoplasma gondii Histone H2A1 on Murine Macrophages and Encapsulation with Polymer as a Vaccine Candidate"

_vaccines, 2020, doi:10.3390/vaccines8040731_

Round 1

Reviewer 1 Report

The Authors greatly improved the manuscript by adding novel data and statistical analysis. However, some points required to be further considered in order to clarify the study design and results, and to enhance its scientific soundness.

The introduction should be shortened and clarified, being the aim of the study not focused. Particularly, lines 44-69 should be rewritten in order to better focus the background of the study.

I can propose the following structure:

  1. “An inhibition of T. gondii infection has been found associated with the release of IFN-y in many cell types. Particularly, anti-T. gondii effects occurred in macrophages pre-exposed to IFN-γ, but not to T. gondii first and then to IFN-γ. Similarly, an increased secretion of reactive oxygen species (ROS) by macrophages, due to lacking NADPH oxidase, has been correlated with the inhibition of T. gondii.”
  2. Role of inflammation in the anti-T. gondii activity
  3. Role of histones in T. gondii replication.
  4. Aim of the study: recombinant TgH2A1 as a modulator of macrophage response and immune system, and delivery of rTgH2A1 through encapsulation in biodegradable PLGA and chitosan nanospheres to improve these properties. Details about delivery systems can be deleted in the introduction and included in the discussion.

According to the changes made in the Introduction, discussion could be improved.

Further comments

Figure 1: please explain (a)

Lines 313-315: the wording should be improved

Figure 5b: regarding the NO levels, the quantification method should be detailed.

Several typing inaccuracies (for instance, the spaces before parenthesis) and grammar errors require to be corrected. This reviewer believes that the help of an English mother tongue could improve the English style thus making the manuscript content more fluent and immediate.

Reviewer 2 Report

The article by ZhengQing Yu and colleagues describes experiments to generate recombinant Toxoplasma gondii (Tg) histone H2A1 and use it to vaccinate mice against Tg infection. The strengths of the paper are the originality and the scope of the investigation. The introduction is sufficiently informative and appears to be well-referenced. After reading the paper, I was convinced that this is a vaccination strategy worthy of further examination. The quality of the paper is unfortunately hampered by poor use of the English language. Nearly every sentence could benefit from reorganization, improved word choice, and/or removal of extraneous words. This impedes the comprehension of the paper and detracts significantly from its merits. Notable instances of language issues that demand further clarification are outlined in the minor issues below.

Major issues:

  1. The first paragraph of section 2.6 must make it clear that the sera from the rTgH2A1 immunized rats is different from the T. gondii-infected rats’ sera, as this is a critical part of the method, and data from this sera was used to validate the protein product. Instead, on my first read-through I thought that the sentence from line 155 was a continuation of the procedure for obtaining rTgH2A1-specific sera. Consider splitting this into two paragraphs to improve clarity.
  2. Data is missing for section 3.1. A southern blot and the sequence alignment would be appropriate here as, at minimum, supplemental data.
  3. There are multiple issues with the parts of Figure 1.
    1. Fig 1a please substitute a cropped original gel image from that in the unpublished data files rather than this one that has clearly been cut and pasted together. Also, the notation “(a)” is not included in the caption for Figure 1.
    2. From the unpublished figures it appears evident that the protein ladder used for the SDS-PAGE and the Western blot are the same, which is good and ideal. However, the ladders are annotated differently between the SDS-PAGE and the Western blot. The 10 kDa band for lanes 1 of (b) and (c) appears to have run off the gel (and therefore should not be labelled in the figures for (b) and (c), but it is visible in the control gel that was used for the lanes “2”. The band annotations should be made consistent between (a), (b) and (c). If indeed different protein standard ladders were used, the specific product name should be provided.
  4. Line 329 refers to a Table S1, but this was not provided in the materials for me to review.
  5. In the Methods and Materials, it states that “the levels of IgG1 and IgG2a … were measured by using the double antibody sandwich ELISA kits… referenced to known concentrations of recombinant mouse IgG1, IgG2a…” Therefore, please include the calculated concentrations of IgG1 and IgG2a in Table 2 in addition to raw OD values. This is important to make your point that there is more IgG2a than IgG1 in the serum, as OD values are not directly informative of concentration. The ratio of IgG2a versus IgG1 in serum seems similar between rTgH2A1/PLGA and rTgH2A1/chitosan nanoparticles in Table 2. However, the text asserts that the rTgH2A1/PLGA nanoparticles cause more of a Th2 skew than the rTgH2A1/chitosan. Revealing the calculated concentrations of these antibody subtypes would have bearing on this argument.
  6. Use of the word “protect” throughout the results and conclusion is overly strong considering that all mice still succumbed to the infection. Consider modulating to “prolonged survival” or similar.

Minor Comments:

Line 21, 77: “with high security” Do the authors mean “with a strong safety profile”?

Line 107: Obtained by immortalization of murine macrophages from C57BL/6 mice…

Line 127 “targeting the 573 bp”: either here or in line 120 the authors should specify that 573 bp is the length of the H2A1 ORF

Line 130: “into the same sites of the pET32a vector” Same as what?

Line 151: What is the site of subcutaneous injection?

Line 160: Please include product name; e.g. Trans-Blot system

Line 188: Cell Counting Kit-8 (CCK-8, Beyotime, Shanhai, China) - introduce the abbreviation CCK-8 (line 192) before using it.

Line 320: “cells were treated with PBS and purified His-tagged protein, respectively, while the experimental group was treated…”

Figure 4: Should be annotated in three parts, (a), (b) and (c) and labelled as such in the figure and caption.

Lines 348-349: no significant differences (in what assay?) were observed between the blank and control groups? The reader should not have to read the next several sentences to comprehend the first sentence of the paragraph.

Line 361: “**P<0.01” can be omitted as this notation is not used in the figure.

Line 428: include “in week 4”

Line 520: The average diameter of chitosan nanospheres is missing from this sentence.

Line 575: The survival rate of mice is considered the best way to evaluate T. gondii vaccine efficiency. (Ref 90)

I cannot find reference 51 used in the body of the paper. If it is unnecessary, this self-citation should be removed.

Round 2

Reviewer 1 Report

The Authors have improved the article according to the reviewer comments, especially regarding the Introduction (lines 45-67).

I can further suggest to modify the last sentence of Introduction (lines 80-82) "In the in vivo environment, the mice immunized with rTgH2A1 encapsulated in biodegradable poly (DL-lactide-co-glycolide) (PLGA) and chitosan nanospheres caused a prolonged survival time" as follows

"Moreover, immunomodulation of rTgH2A1 encapsulated in biodegradable poly (DL-lactide-co-glycolide) (PLGA) and chitosan nanospheres have been evaluated in mice model".

Reviewer 2 Report

I appreciate the significant improvement the authors have made to the manuscript. It is clearly improved over the previous submission. The English language usage is better, but could still stand to be improved with editorial assistance. More importantly, there are still some problems with the paper's conclusions.

Major issues:

1.)  Line 37: your reference (1) states: "In northern European countries the seroprevalence can be as low as 10%, whereas in some areas of Brazil [39] and in Madagascar [40] it can be as high as 80%. In the USA, Jones et al. estimated that the overall age-adjusted seroprevalence of T. gondii infection is 11% [23]." I am not seeing where the authors obtained the data for "causes infections in approximately 30% of the population". Please provide an additional reference, explain your calculation, or edit this value.

2.) Line 501: "and that inhibited the replication of T. gondii" was not demonstrated in this paper. Please either remove this assertion, or modulate it to "is expected to inhibit the replication..." and include an appropriate reference."

3.) Most importantly, the data does not support the assertion in line 540-541 that rTgH2A1/PLGA nanospheres generate a slightly stronger Th1 oriented immune response than rTgH2A1/chitosan nanospheres. A simple ratio of IgG2a/IgG1 for each type of nanosphere reveals that the ratio of IgG2a to IgG1 is equivalent between the two formulations to 4 decimal places. Please remove any suggestion of this difference from the manuscript.

Minor comments:

1.) Line 50-52 needs fixing. The grammar is incorrect and the subject matter does not flow clearly from the rest of the paragraph.

2.) Line 62 is missing the word "cytokines".

3.) Line 118: I see that you tried to address my previous comment here, but I would still like to see "targeting the complete 543 bp H2A1 ORF..."

4.) Line 409-411: Revise this sentence to refer to the calculated concentrations rather than the OD values. Remove the word "much", and change "every group" to "the rTgH2A1/nanosphere vaccinated groups".

5.) Line 483: The results which were significant at 40 micrograms/mL should be specifically mentioned prior to the comparison with the 80 micrograms/mL dose, either in the same sentence or earlier in the paragraph.

6.) Line 512: Please change the phrase "with high security". That does not translate.

Author Response

This manuscript is a resubmission of an earlier submission. The following is a list of the peer review reports and author responses from that submission.

Round 1

Reviewer 1 Report

Present study is focused on the evaluation of the ability of a recombinant tTgH2A1 to modulate the immune response mediated by macrophages. Moreover, the authors developed two type of possible nanospheres in order to protect tTgH2A1 from degradation, thus improving its activity.

The Authors described both the development of tTgH2A1 and nanocarriers and the evaluation of the immunomodulatory activity in macrophages and in an in vivo mice model o T. gondii infection.

Results highlighted a potential interest for tTgH2A1 and its formulations with PLGA and chitosan for the vaccination against T. gondii infection.

The study appears to be of interest in the field; however, some points require to be improved before publication.

  • Introduction is too long and knowledge about T. gondii can be avoided in order to better focus the aim of the study, which is to evaluate the ability of a recombinant T. gondii H2A1 (rTgH2A1) both alone and encapsulated in PLGA nanospheres to modulate the immune response of macrophages.
  • Lines 56-57: the wording can be deleted in the introduction. Conversely, it is more appropriate in the discussion to support the obtained results.
  • The interest in nanospheres should be better contextualized in the introduction.
  • The choice of the administration route and dosages in in vivo study should be justified.
  • English style should be revised thoroughly in all the manuscript and some wordings (see for instance lines 40-41, 120-123, 319, 327, 527, 538…) require to be clarified.
  • Formatting inaccuracies should be corrected
  • Figure 1. kDA scales should be improved so that the displayed number should be clearly read
  • The title of paragraph 4 rTgH2A1 affects the proliferation of murine macrophages, induced apoptosis in not clear. Please, check and correct
  • Bar graphs in figure 4 require to be improved for more clarity. The bars representing the same treatment at different concentrations can be in the same style.
  • In all the figures, the statistical analysis method should be reported.
  • Lines 365-366: the wording can be deleted
  • Figure 6c: the different behaviour of 20 and 80 μg/ml of rTgH2A1 should be explained.
  • Please, correct ug/ml using the suitable symbols
  • Figure 7: please, improve the figure quality
  • Figure 8b: a very low difference in the OD of rTgH2A1/PLGA compared to the blank is displayed. Please, check the statistical analysis.
  • Discussion can be improved by focusing on the obtained results. Some knowledge already described in the introduction can be deleted.

Reviewer 2 Report

In this paper the authors evaluate the immune response (total IgG and IgG subclasses, splenocyte proliferation and seric cytokine secretion) and protection after vaccination with Toxoplasma H2A1 protein encapsulated in PLGA and chitosan nanoparticles. They also look at the in vitro activation of a macrophage cell line by this protein.

The work is relatively well designed, however the question is not original and the results provide few advance in current knowledge. The paper will be of interest only to a limited number of people. There is not enough

The aim of the study especially the choice of the protein used for immunization and the way of challenge are not enough explain and justified. The protein is not described at all because the authors want to publish the description of the protein in another publication. The methods require in some sections clarification and a little more detail and several points need to be discussed.

Minor corrections :

Materials and methods :

The Ana1 cell line characteristics are not given.

The particles are not well characterized. The particles’ average hydrodynamic diameters and the ζ potentials are not measured. Moreover, the scanning electron microscopy images of the particles are not very indicative.

The design of the in vivo experiment is difficult to understand. The authors have 15 mice/group but they use only 5 mice to get sera and 10 mice are used for the challenge. In my opinion, from an ethical and to get more robust results they could used only 10 mice and get the sera from the same mice rather than use 5 other mice to analyze the humoral response and cytokine production.

The authors used the Endotoxin Removal Kit, however, they do not precised if they have quantified the endotoxin level even after the endotoxin removal.

Figures :

Fig 1a is dispensable since this is a classical cloning technic. For c and d the control with the sera from naive rat is not on the same blott and seems to be done in a different way. The same is true for the native protein. It should be better to have all the proteins on the same blott.

Fig 2 : there is not relevant negative and/or positive controls. A irrelevant control recombinant protein or the native protein should be added in this figure.

Major

The description of the H2A1 protein and its interest as a vaccine target need to be justify. The authors explained that this protein is secreted but they dont explain why because this is a nuclear protein. They don’t explain why this protein is supposed to bind to the cell line membrane.

Overall the justification of the choice of the protein to vaccinate the mice need to be explain.

The challenge is performed using tachyzoites injected intraperitoneally. This way of injection and this form of parasite are not really relevant to a natural infection. In humans and target animals, Toxoplasma infection occur through cysts or sporozoites ingestion of type II strain. It will be more appropriate to used oral infection with cysts and to see reduction of parasite load in the tissues.

Moreover from an ethical point of view the survival time for a pathology that do not induce death in the target host may be discutable. The choice of the way of infection and the strain should be discussed and justified.

Overall I suggest that a more relevant publication combining the data concerning the description of the protein in the excretory/secretory antigens and also may be the results the authors already have on the vaccine trials with a more relevant type of challenge (by example oral challenge with cysts) against chronic toxoplasmosis.